# Change in childbearing intention, use of contraception, unwanted pregnancies, and related adverse events during the COVID-19 pandemic: Results from a panel study in rural Burkina Faso

**Thomas Druetz**[1,2,3]*, **Sarah Cooper**[1], **Frank Bicaba**[4], **Alice Bila**[4], **Martine Shareck**[5], **David-Martin Milot**[5], **Cheick Tiendrebeogo**[1], **Abel Bicaba**[4]

**1** Département de médecine sociale et préventive, École de santé publique, Université de Montréal, Montréal, Canada, **2** Centre de recherche en santé publique, Montréal, Canada, **3** Center for Applied Malaria Research and Evaluation, Tulane University, New Orleans, Louisiana, United States of America, **4** Société d'Études et de Recherches en Santé Publique, Ouagadougou, Burkina Faso, **5** Département des sciences de la santé communautaire, Université de Sherbrooke, Sherbrooke, Canada

* thomas.druetz@umontreal.ca

**Data Availability Statement:** We do not have consent from participants to share data unless it is

## Abstract

Evidence on how the COVID-19 pandemic has affected women's reproductive health remains scarce, particularly for low- and middle-income countries. Deleterious indirect effects seem likely, particularly on access to contraception and risk of unwanted pregnancies, but rigorous evaluations using quasi-experimental designs are lacking. Taking a diachronic perspective, we aimed to investigate the effects of the pandemic on four indicators of women's reproductive health: history of recent adverse events during pregnancy (past), use of contraception and unwanted pregnancies (present), and childbearing intentions (future). This study was conducted in four rural health districts of Burkina Faso: Banfora, Leo, Sindou and Tenado. Two rounds of household surveys (before and during the pandemic) were conducted in a panel of 696 households using standardized questionnaires. The households were selected using a stratified two-stage random sampling method. All women aged 15–49 years living in the household were eligible for the study. The same households were visited twice, in February 2020 and February 2021. The effects were estimated by fitting hierarchical regression models with fixed effects or random intercepts at the individual level. A total of 814 and 597 women reported being sexually active before and during the COVID-19 pandemic, respectively. The odds of not wanting (any more) children were two times higher during the pandemic than before (2.0, 95% CI [1.32–3.04]). Among those with childbearing intention, the average desired delay until the next pregnancy increased from 28.7 to 32.8 months. When comparing 2021 versus 2020, there was an increase in the adjusted odds ratio of contraception use (1.23, 95% CI [1.08–1.40]), unwanted pregnancies (2.07, 95% CI [1.01–4.25]), and self-reported history of miscarriages, abortions, or stillbirths in the previous 12 months (2.4, 95% CI [1.04–5.43]). Our findings in rural Burkina Faso do not support the predicted detrimental effects of COVID-19 on

for the purpose of a study approved by a research ethics committee. These restrictions were established by the Comité d'éthique de la recherche en sciences de la santé at University of Montreal. Justified requests for access to anonymized data and Stata scripts used for this analysis can be sent to the data trustee at gregory.moullec@umontreal.ca.

**Funding:** This work was carried out with the aid of a grant (#108553) awarded to TD from the Innovating for Maternal and Child Health in Africa initiative—a partnership of Global Affairs Canada (GAC), the Canadian Institutes of Health Research (CIHR) and Canada's International Development Research Centre (IDRC). It also received support from the Réseau de la Recherche en Santé Publique du Québec (RRSPQ). TD is a FRQS Junior 1 Research Scholar. MS holds a Tier 2 Canada Research Chair in Urban Health Equity and Young People. The funders had no role in study design, data collection and analysis, decision to publish, or preparation of the manuscript.

**Competing interests:** The authors have declared that no competing interests exist.

**Abbreviations:** CI, Confidence Interval; DHS, Demographic and health surveys; LMICs, Low- and middle-income countries; OR, Odds ratio.

the use of family planning services in LMICs, but confirm that it negatively affects pregnancy intentions. Use of contraception increased significantly among women in the panel, but arguably not enough to avoid an increase in unwanted pregnancies.

## Introduction

The COVID-19 pandemic has affected the lives and prospects of billions of individuals and is likely to have a significant impact on global demographics beyond the directly attributable mortality burden. Notably, it has been hypothesized that the COVID-19 pandemic may affect reproductive health, a phenomenon that has been observed during previous social, economic, and health crises [1].

In the short term, pandemics and the pathogens that cause them can give rise to a rapid decline in natality due to an increase in adverse events during gestation and/or a decrease in conception. A reduction in the population and higher rates of maternal mortality and still-births were contributory factors to the decreased in birth rate in the USA during the 1918 influenza pandemic [2]. More recently, in Sierra Leone, the 2013–2016 Ebola outbreak indi-rectly caused >1400 maternal deaths or stillbirths resulting from disruptions and decreased utilization of routine health services [3]. Economic, social, humanitarian, and health crises can also reduce the desire for conception or postpone the decision to have a child [4–6]. This can result from various psychological mechanisms, including uncertainty, stress, concern about the future, fear of infection and/or vertical transmission during conception or pregnancy, social isolation, and spousal separation, among other factors [7]. Some evidence suggests that the current pandemic has been associated with a decrease in fertility plans in several countries, including the USA, Italy, and the United Kingdom [8, 9].

However, several factors related to pandemics can have the opposite effect and increase birth rates. In particular, health system disruptions can reduce access to contraception and family planning clinics, as was observed during the Ebola outbreak in West Africa [10]. Crises and stressful environments or events are associated with increased rates of forced marriage, sexual and gender-based violence, sexual exploitation, and a decrease in women's reproductive autonomy [9, 11]. Since the onset of the COVID-19 pandemic, there have been reports in dif-ferent settings (high-, middle-, and low-income countries) of increased difficulty in obtaining reproductive health services because of lockdowns and supply chain issues for contraception commodities [12, 13]. An increase in gender-based and sexual violence has also been reported in many countries, including France, the UK, Kenya, the USA, Singapore, and Argentina [11, 14, 15]. This can lead to a greater number of pregnancies, especially unwanted pregnancies [16].

With this range of possible repercussions, it is unclear how reproductive health indicators will be affected during the COVID-19 pandemic. Most predictions anticipate an increase in birth rates in low- and middle-income countries (LMICs) due to shortage and inaccessibility of contraceptive services [1, 12, 13, 17, 18]. However, analyzing the situation in LMICs as a sin-gle entity is oversimplified and misleading. Even within countries or regions, the outbreak and the preventive or mitigation measures are significantly heterogeneous. Furthermore, their effects will likely vary on multiple axes, including geographical area, time, socioeconomic sta-tus, and the degree of women's empowerment. To fully understand the ramifications of the pandemic, local investigations must be conducted in restricted study areas using an intersec-tional lens [19].

Empirical evidence on the topic is scarce. A phone-based survey conducted in four sub-Saharan African countries in May–July 2020 did not find a deleterious effect of the pandemic on family planning services. In contrast, a study on administrative data in Mozambique found a modest but short-term and transient reduction in contraceptive use in the three months following March 2020 [20, 21]. Other important sexual and reproductive health indicators remain to be examined, such as conception planning, unwanted pregnancies, and pregnancies that did not end in a live birth.

This study aims to investigate the effects of the COVID-19 pandemic on four reproductive health indicators: desire for another child; preferred lapse of time until subsequent pregnancy; use of contraception; and history of miscarriages, abortion, or stillbirth in the previous 12 months. We conducted repeated household surveys before and during the pandemic in a panel of 700 randomly selected households in rural Burkina Faso. We hypothesized that the pandemic would reduce women's desire for another child or delay their intention to become pregnant. An increase in unmet need for contraception was also anticipated, based on the assumption that supply would not meet the rising demand. Consequently, the prevalence of women with a recent history of abortion was expected to increase, and no effects on miscarriages and stillbirths were anticipated. By examining these different hypotheses, this study is the first to offer an integrated perspective on recent past events, current practices, and future intentions regarding women's reproductive health and how they have been altered during the COVID-19 pandemic.

## Methods

### Ethics statement

All participants recruited in 2020 provided written informed consent. As suggested and approved by the research ethics committees, all participants recruited in 2021 provided informed verbal consent, in order to reduce risk of COVID-19 transmission. Verbal consent was recorded by the surveyor on the mobile data collection platform. The questionnaire was administered individually in a secluded area to preserve participant confidentiality. Participants aged 15–17 years old were considered mature minors and consented as adults. All the study procedures, including those for obtaining consent, were approved by the Comité d'éthique de la recherche en sciences de la santé at University of Montreal (Certificate #CERSES-20-146-D) and by the Comité d'éthique pour la Recherche en Santé in Burkina Faso (Deliberation #2018-6-075). The funder of the study had no role in study design, data collection, data analysis, data interpretation, or writing of the manuscript.

### Burkina Faso in the context of COVID-19

Burkina Faso is a landlocked Sahelian country with a total population of ~20 million. It ranked 182 out of 189 countries on the 2020 Human Development Index [22]. Few deaths have been directly attributable to COVID-19 in Burkina Faso (169 as of late July 2021), and the cumulative number of confirmed cases is under 15,000. However, due to limited testing and surveillance capacity, the burden is likely underestimated [23]. By measuring the prevalence of antibodies, seroprevalence surveys have confirmed that transmission is significantly higher than what would be expected from the surveillance data [24].

Authorities in Burkina Faso enforced several measures intermittently throughout 2020, including a ban on mass gatherings, curfews, quarantine of cities, travel restrictions within the country, lockdown, and closure of borders. The second wave of COVID-19 was the largest, with a peak of ~250–300 new daily cases in December 2020 and January 2021. Officially, the provision of maternal and reproductive health services has not been altered as a result of the

pandemic. As part of their pandemic preparedness plan, health authorities stockpiled supplies and drugs (including contraceptives) in early 2020 in anticipation of an increased risk of supply chain disruptions [20]. However, a phone-based survey conducted in the health facilities of two large cities in Burkina Faso, including Ouagadougou, suggested numerous disruptions in the provision of maternal and child health services [25]. Notably, more than 50% of health facility providers reported that they had temporarily suspended family planning services during the pandemic.

## Study design and participants

We conducted a quasi-experimental study nested in a panel study (SYNERGIE) which aims to assess the implementation and effects of universal healthcare policies in rural and semi-rural Burkina Faso [26]. The study area comprised four health districts (Banfora, Sindou, Leo, and Tenado) purposively selected based on security and location (rural settings with only small towns) [27]. The study area has a total estimated population of 988,447 within an area of 21,472 km$^2$ (Fig 1) [28]. All study parameters were originally decided based on the SYNERGIE study protocol; the present study is a natural experiment that exploited the already existent research platform to evaluate the effects of an unanticipated event, i.e. the occurrence of the pandemic [29].

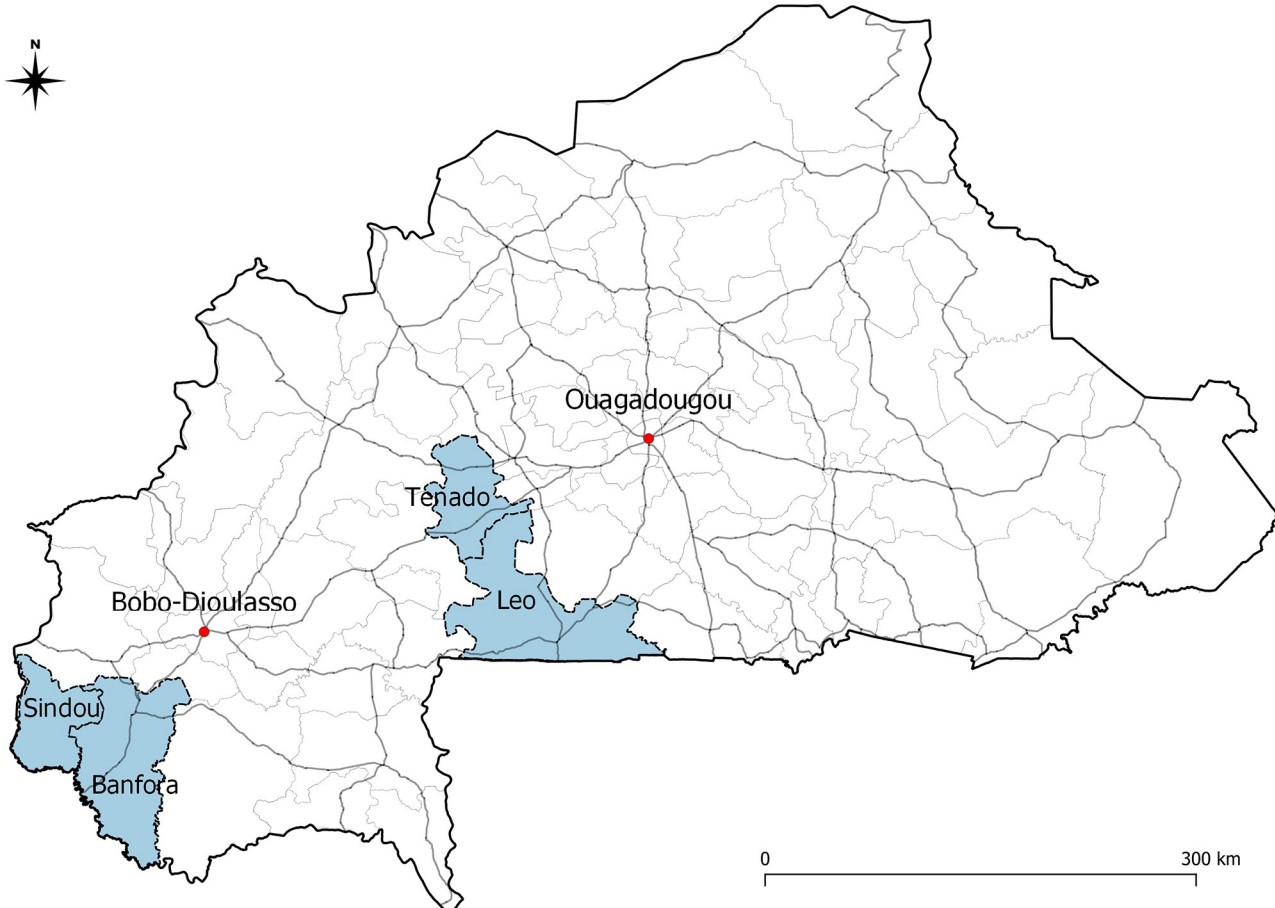

**Fig 1. Map of study area.** The four health districts are displayed in blue. Main roads are shown as gray lines. Data and layers used to create this map are available at: https://data.humdata.org/dataset/burkina-faso-administrative-boundaries.

Two rounds of survey data were collected among the same panel of households in February 2020, prior to COVID-19, and February 2021, during the descending curve of the largest wave of COVID-19 transmission in the country. The households were sampled using a stratified two-stage random sampling method, per the Demographic and Health Survey (DHS) protocol [30]. Among the enumeration areas in the four districts, 29 were first selected with probability proportional to size. In a second stage, 24 households per enumeration area were randomly selected from a complete listing of households. Households without women of reproductive age were not eligible and were replaced by the nearest eligible one. The target sample size was decided for the purpose of another study, and was set to a maximum of 700 households. All households recruited in 2020 were systematically visited again the next year; losses to follow-up were not replaced.

All women aged 15–49 years and living in the sampled households were eligible and invited to participate in the study during the door-to-door survey. The nested cohort of women was open and dynamic (i.e., all women surveyed in 2020 remained eligible the following year); therefore, in 2021, all women aged 15–50 years were invited to participate. For logistical reasons, households were only visited once per survey round unless all eligible women were temporarily absent, in which case a follow-up visit was scheduled.

## Survey procedures

A standardized sociodemographic questionnaire was administered individually to all participants. Questions extracted from the DHS instruments gathered data on household features (e.g., size, location, type of construction, possessions, etc.) which were used to create a socioeconomic index as per the DHS guidelines [30]. Using the DHS standardized questions (available at: https://dhsprogram.com/publications/publication-dhsq7-dhs-questionnaires-and-manuals.cfm), women were also asked about their reproductive health: history of live births and miscarriages, pregnancy status, childbearing intentions, and use of contraceptives. Questions regarding knowledge and perceptions towards COVID-19 were added for the 2021 round (S1 Text).

Surveys were administered at the participants' homes by female research assistants who received special training in epidemiological research. Questionnaires were completed on a mobile data collection platform (CommCare, Dimagi, Cambridge, MA) installed on Android tablets. Questionnaire data were automatically pushed to a secure cloud-based server using the local mobile phone network. Most (>80%) of the female research assistants were hired for both survey rounds. Participants answered the questions in the language of their choice (French, Moore, Dioula, Fulfuldé, or Gourmantché).

## Outcome definitions

All analyses are limited to women who reported being sexually active. Several outcomes were defined to better delineate and understand the hypothesized changes in reproductive health practices since the onset of the pandemic. The first study outcome is women's desire (yes/no, binary variable) to have children, or another child, in the future. The second outcome is the length of time, in months, that women would prefer to wait before their subsequent pregnancy (continuous variable); this analysis is restricted to those women who declared that they wanted children in the future. The third outcome is the actual reproductive status of the women at the time of the survey, with four mutually exclusive categories (categorical variable): (i) pregnant (wanted/planned); (ii) pregnant (unwanted); (iii) non-pregnant and using contraception; (iv) non-pregnant and not using contraception. Contraception use was defined as currently taking at least one measure recommended at the health facility to avoid pregnancy (e.g., injectable,

implants, pills, intrauterine devices). The fourth outcome was the self-reported history (yes/no, binary variable) in the last 12 months of a pregnancy that did not end in a live birth, disregarding the reason (abortion, miscarriage, or stillbirth).

## Data analysis

The panel composition was compared (before vs. during pandemic) using Pearson's $\chi^2$ tests. Variations in the study outcomes over the one-year period were assessed by the coefficient of a binary exposure variable, while the longitudinal data structure was exploited by yearly nesting observations within participants. Models were fitted using logistic, multinomial, or linear distribution, depending on the type of study outcome (binary, categorical, or continuous, respectively). Fixed effects were included in the logistic models to isolate changes only attributable to time-varying factors within individuals. This approach was not possible for the linear and multinomial models due to convergence issues caused by the smaller number of participants included in these analyses; instead, mixed effects models with random intercepts at the individual and district levels were used. Robust variance estimators were consistently used [31].

Covariates tested in the linear and multinomial models included characteristics of the participants (age, parity, education level, marital status, occupation) and the households (socioeconomic status, size, distance to the nearest health facility, rural/urban setting). Multicollinearity was ruled out by verifying that variance inflation factors did not exceed 4. The best-fitting model was selected according to the Akaike Information criterion values. Sensitivity analyses were performed by stratifying by age category, with three categories (15–25, 26–35, >35) defined *a priori*. All analyses were performed using Stata version 14.0 software (StataCorp LLC, College Station, Texas). Maps were produced using QGIS version 3.8.1 Zanzibar (open-source software with general public license).

## Results

Among the 697 households sampled in February 2020, 72% were surveyed again 12 months later (n = 504); the other households were visited but were empty, or had moved outside of the study area and were not replaced. A total of 814 and 597 sexually active women of reproductive age (15–50 years) were recruited in these households in February 2020 and February 2021, respectively. Among these, 540 women were surveyed both years, which represents 77% (n = 1,080) of the 1,411 year-observations included in the analyses. The sociodemographic characteristics of the participants are displayed in Table 1. In 2021, 98.5% of the participants knew what COVID-19 was, and 88.5% thought it was moderately or very threatening.

Compared with the pre-pandemic situation, women had a 33% increase in the likelihood of not wanting children/another child (RR = 1.33; 95% CI [1.16–1.53]). Although the increase was observed in all age categories, it was only statistically significant in women aged 26–35 (RR = 1.5; 95% CI [1.19–1.89]) (Fig 2). Of note, among those who wanted to have children, the desired lapse of time until the next pregnancy increased between 2020 and 2021 (Fig 3). Indeed, the average desired delay rose significantly from 31.6 to 36.7 months among 15–25-year-olds (diff.: +5.09, 95% CI [0.93–9.28]). The average delay also rose from 30.8 to 34.5 months and from 18.3 to 21.4 months among those aged 26–35 and ≥35 years, respectively, although the increase was not statistically significant.

Women's actual reproductive status differed significantly before and during the pandemic. Based on their pregnancy status and self-reported use of contraception, four categories were defined: (i) non-pregnant and using contraception, (ii) non-pregnant and not using contraception, (iii) with desired pregnancy, and (iv) with unwanted pregnancy. The model indicates significant changes in the 2021 age-adjusted distribution compared to the pre-pandemic

**Table 1. Sociodemographic characteristics of sexually active women in the four health districts under study, in February 2020 and 2021.**

| | All surveyed women | | | Women surveyed both years | | |
|---|---|---|---|---|---|---|
| | February 2020 | February 2021 | Chi$^2$ test | February 2020 | February 2021 | Chi$^2$-test |
| | n (%) | n (%) | P-value | n (%) | n (%) | P-value |
| Age group | | | | | | |
| 16–25 y | 252 (31.3) | 152 (25.9) | 0.016 | 149 (27.7) | 129 (23.8) | 0.250 |
| 26–35 y | 291 (36.2) | 204 (34.7) | | 199 (37.0) | 188 (34.7) | |
| >35 y | 262 (32.55) | 232 (39.5) | | 188 (34.6) | 219 (40.8) | |
| Went to primary school | | | | | | |
| Yes | 355 (41.2) | 243 (40.7) | 0.865 | 210 (39.0) | 216 (39.8) | 0.709 |
| No | 479 (58.9) | 354 (59.3) | | 330 (60.9) | 324 (60.2) | |
| Motherhood | | | | | | |
| Nulliparous | 101 (12.4) | 70 (11.8) | 0.269 | 54 (10.0) | 55 (10.5) | 0.901 |
| Primiparous | 123 (15.1) | 73 (12.3) | | 68 (12.7) | 66 (11.9) | |
| Multiparous | 590 (72.5) | 451 (75.9) | | 418 (77.3) | 417 (77.5) | |
| Pregnant | | | | | | |
| Yes | 78 (9.6) | 63 (10.5) | 0.819 | 52 (9.7) | 56 (10.3) | 0.208 |
| No | 732 (90.4) | 534 (89.4) | | 485 (89.9) | 484 (89.7) | |
| Contraception use | | | | | | |
| Yes | 329 (40.4) | 279 (46.7) | 0.018 | 227 (42.2) | 260 (48.0) | 0.043 |
| No | 485 (59.6) | 318 (53.3) | | 313 (57.8) | 280 (51.9) | |
| Married or in cohabitation | | | | | | |
| Yes | 634 (77.9) | 471 (78.9) | 0.650 | 442 (81.7) | 438 (81.2) | 0.754 |
| No | 180 (22.1) | 126 (21.1) | | 98 (18.2) | 102 (18.8) | |
| Received a salary in the last month | | | | | | |
| Yes | 396 (48.7) | 297 (49.8) | 0.683 | 272 (50.1) | 271 (50.3) | 0.952 |
| No | 418 (51.3) | 300 (50.2) | | 268 (49.8) | 269 (49.6) | |
| Knowledge about COVID19 | | | | | | |
| Yes | | 588 (98%) | NA | | 534 (98%) | NA |
| Size of household | | | | | | |
| 0–5 | 234 (28.8) | 209 (35.1) | <0.001 | 148 (27.5) | 202 (37.3) | <0.001 |
| 6–10 | 396 (48.7) | 306 (51.3) | | 259 (48.1) | 272 (50.1) | |
| >10 | 183 (22.5) | 82 (13.7) | | 132 (24.3) | 66 (12.5) | |
| Residence | | | | | | |
| Urban | 405 (49.8) | 282 (47.2) | 0.35 | 254 (46.9) | 254 (46.9) | 1 |
| Rural | 409 (50.2) | 315 (52.8) | | 286 (53.0) | 286 (53.0) | |
| District | | | | | | |
| Banfora | 457 (56.1) | 338 (56.6) | 0.357 | 294 (54.3) | 294 (54.3) | 1 |
| Tenado | 111 (13.6) | 96 (16.1) | | 84 (15.6) | 84 (15.6) | |
| Leo | 61 (7.5) | 47 (7.9) | | 48 (8.9) | 48 (8.9) | |
| Sindou | 185 (22.7) | 116 (19.4) | | 114 (21.1) | 114 (21.1) | |

situation (Fig 4). The likelihood of not using contraception while non-pregnant significantly decreased in 2021 compared to 2020 (RR = 0.83, 95% CI [0.73–0.93]) (Table 2), contrary to the likelihood of using contraception (RR = 1.23, 95% CI [1.08–1.40]). The likelihood of an unwanted pregnancy increased during the pandemic compared to the pre-pandemic period (RR = 2.07, 95% CI [1.01–4.25]). On the other hand, the data did not indicate a change in desired pregnancies (RR = 0.86, 95% CI [0.56–1.32]).

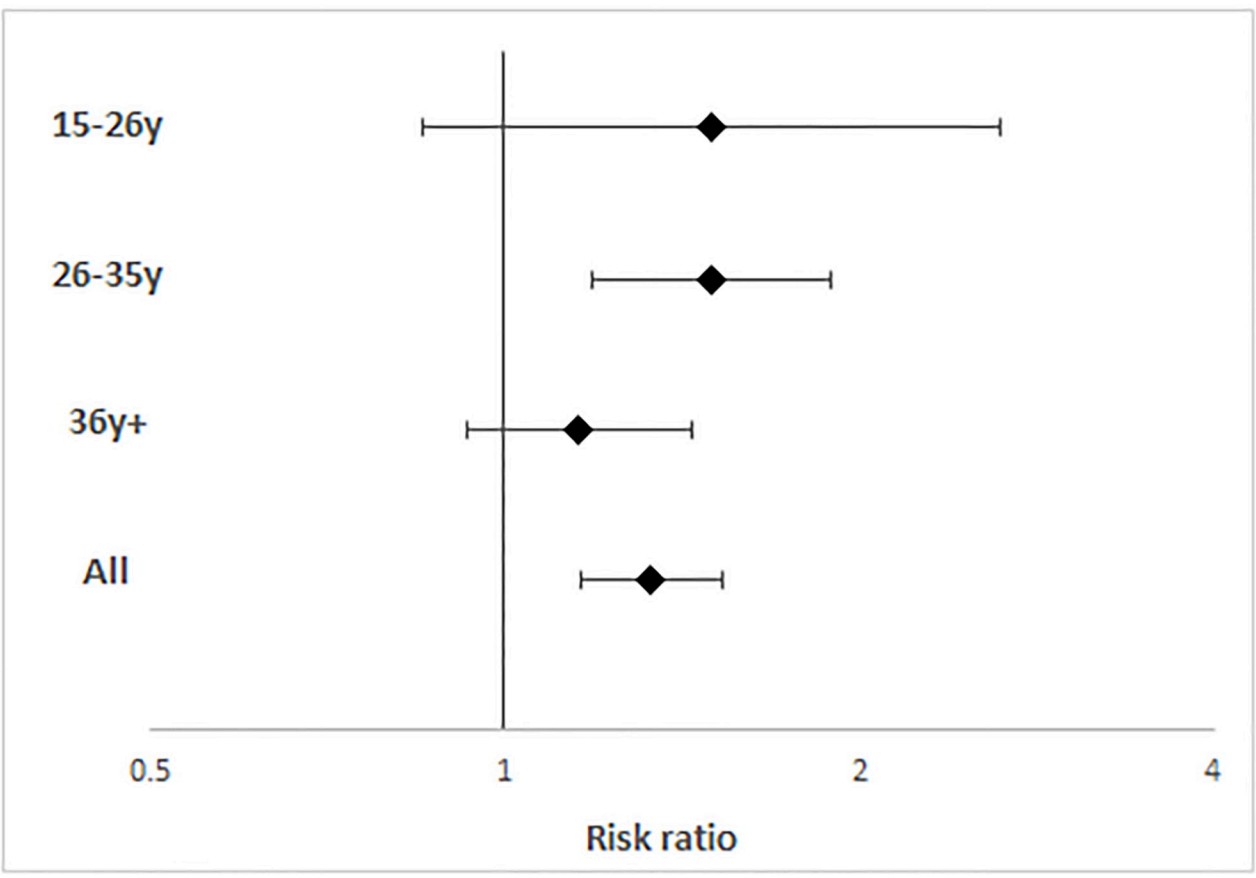

**Fig 2. Desire not to have (more) children in women in the four health districts under study, with risk ratio comparing 2021 vs. 2020 stratified by age category.** Estimates were derived from a logistic regression model with fixed effects at the individual level. Age categories are: 16–25 years, 26–35 years, 36 years and older. The odds ratios with fixed effects are displayed on the x axis using a logarithmic scale. Estimates are presented with their 95% confidence interval.

Finally, the self-reported history of miscarriages, abortions, or stillbirths in the past 12 months significantly increased. When comparing pandemic vs. pre-pandemic survey results, the likelihood for this particular outcome increased by 41% (RR = 1.41, 95% CI [1.10–1.80]) (Table 3).

## Discussion

This study is the first to use repeated surveys in a panel of households to assess changes in reproductive health dynamics during the COVID-19 pandemic in Sub-Saharan Africa. It shows that since the beginning of the COVID-19 pandemic, women living in rural Burkina Faso are more likely to choose not to have any more children or to delay their next pregnancy. It also reveals that the use of contraceptives has increased, but not enough to meet the decline in childbearing intentions. At the same time, the proportion of women with an unwanted pregnancy has increased significantly, as has the risk of having had a recent pregnancy that did not end in a live birth.

Evidence on the impact of critical situations or events on childbearing preferences and demand for contraception is scarce [32]. Some studies suggest that these fertility indicators

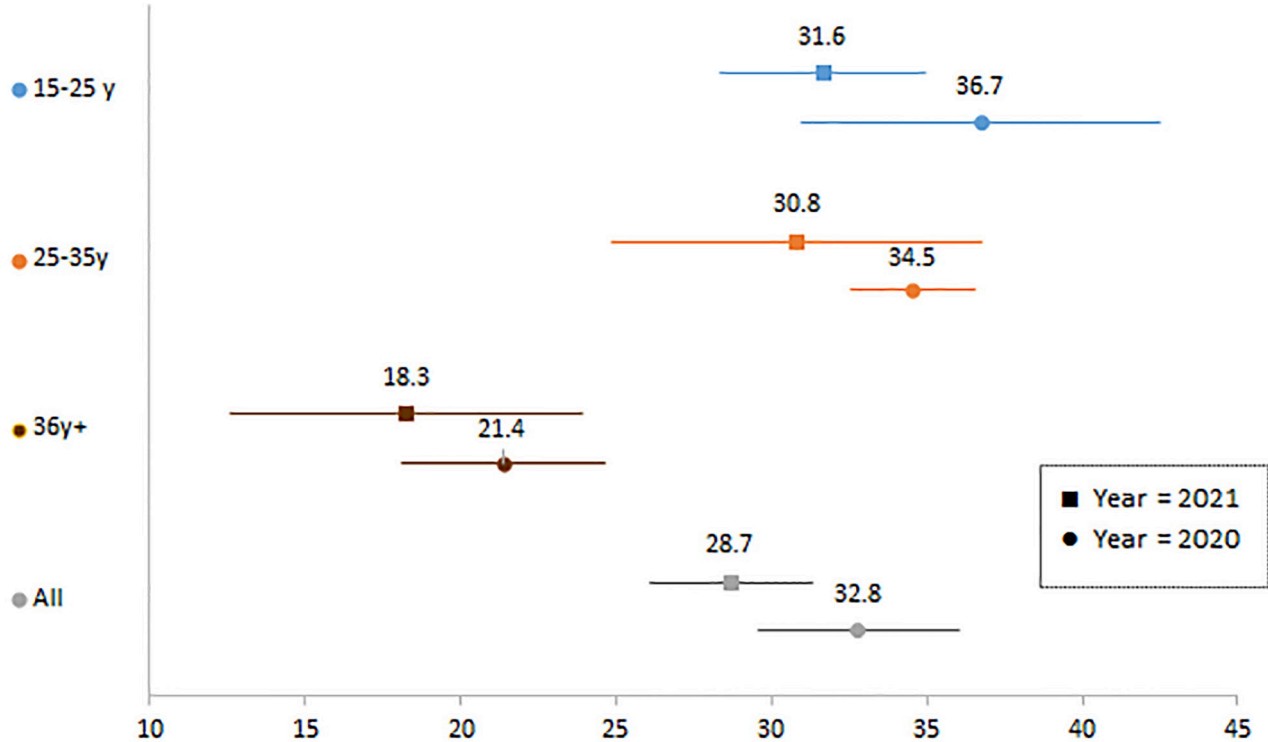

**Fig 3. Desired lapse of time (in months) before next pregnancy in sexually active women in the four health districts under study, by year and age category.** Estimates are derived from a mixed effects regression model, with random intercepts at the individual and district levels. The model is adjusted for the education level of the women and the socioeconomic status of the household. An interaction term between the year and the age category was forced in the model. Results are displayed in color according the age category: 16–25 years in blue, 26–35 years in orange, and 36 years and older in brown. Within each age category, the first (top) estimate represents the year 2020, while the 2021 estimate is displayed below. Estimates are presented with their 95% confidence interval.

can be affected by financial hardship in the household, humanitarian crises, escalations of violence and displacement, or immediate threats to mothers or fetuses, as observed during the Zika epidemic [32–35]. Besides acute crises, studies conducted in various African settings have also noted that uncertainty and fear are driving factors of postponing or avoiding a pregnancy [34, 36, 37]. Our results indicate that COVID-19 has likely had similar effects among our participants. This may seem surprising, knowing that the official burden directly attributable to the COVID-19 appears to be limited in Burkina Faso. However, perception of its severity was high in the panel, as indicated by the 89% of participants who reported that COVID-19 was moderately or very threatening. While the effects of the pandemic on pregnancy intentions have been documented in high-income countries heavily affected by the pandemic, our study indicates that a similar trend may be occurring in countries presumably less impacted [8, 38].

This study's results are not consistent with predictions of decreased access to contraception and a rise in natality rates in LMICs due to the pandemic [1, 12]. On the contrary, the COVID-19 pandemic likely contributed to the increase in contraceptive use in the panel, from 41% in February 2020 to 50% one year later. It is worth noting that Burkina Faso has implemented several interventions to increase access to family planning services over the last few years, notably by removing user fees for contraception and for postnatal visits [39]. The upward trend in contraceptive use has been steady and rapid since 2015; that year, the contraceptive prevalence rates in the Cascades and Centre-Ouest regions (where the study area is

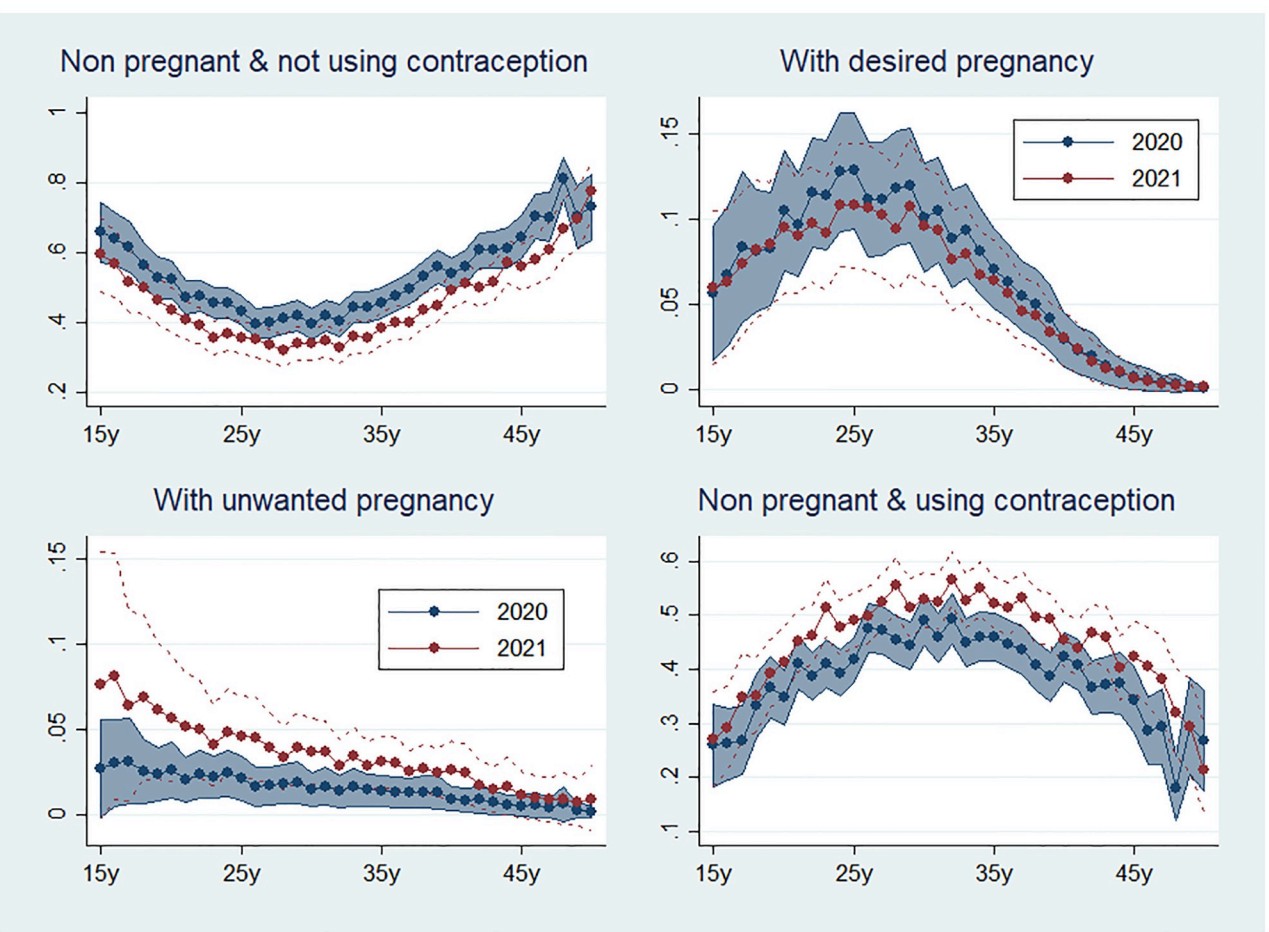

**Fig 4. Women's predicted probabilities in 2020 and 2021 of being (i) non pregnant and not using contraception; (ii) intentionally pregnant; (iii) unintentionally pregnant and; (iv) non pregnant and not using contraception, by age.** Predicted probabilities are derived from a multinomial model adjusted for age, education, parity, marital status, household size, socioeconomic status, area (urban vs. rural) and location of the household, with robust variance estimators. The red line represents the 2021 prediction, with the 95% confidence interval delimited by the red dotted lines. The blue line represents the 2020 prediction, with the 95% confidence interval delimited in the blue-shaded area.

**Table 2. Women's likelihood of being (i) non pregnant and not using contraception; (ii) intentionally pregnant; (iii) unintentionally pregnant and; (iv) non pregnant and not using contraception, with risk ratios comparing 2021 vs. 2020.**

| | Predicted probability | | aRR | 95% CI |
|---|---|---|---|---|
| | **2020** | **2021** | | |
| **Status outcome** | | | | |
| #1. Non pregnant and not using contraception | 0.53 | 0.44 | 0.83 | [0.731–0.934] |
| #2. Non pregnant and using contraception | 0.42 | 0.51 | 1.23 | [1.084–1.395] |
| #3. With desired pregnancy | 0.05 | 0.04 | 0.86 | [0.560–1.321] |
| #4. With unplanned pregnancy | 0.003 | 0.007 | 2.07 | [1.007–4.250] |

aRR adjusted risk ratio; CI Confidence interval

Multinomial model adjusted for age, education, parity, marital status, household size, socioeconomic status, area (urban vs. rural) and location of the household, with robust variance estimators.

**Table 3. Women's likelihood of having had a miscarriage, an abortion, or a stillbirth in the previous 12 months, with risk ratio comparing 2021 vs. 2020.**

| History of miscarriage, abortion or stillbirth | Crude association | | Fixed effects model⁋ | | |
|---|---|---|---|---|---|
| | Predicted probability | Crude RR | RR | 95% CI | P-value |
| 2020 | 0.027 | 1.61 | 1.407 | 1.102–1.798 | 0.006 |
| 2021 | 0.044 | | | | |

RR (adjusted) risk ratio; CI Confidence interval

⁋ Logistic model with fixed effects at the individual level

located) were estimated to be 21% and 28%, respectively, while the national prevalence of unmet need for family planning reached 35% [40–42]. The World Bank estimates suggest a gradual decline in the birth rate in Burkina Faso over the past decade, but it remains one of the highest in the world, with ~37 births per 1,000 inhabitants per year in 2019.

The increase in contraceptive prevalence rates was more modest in our study (+9.2 percentage points) than in a recent phone-based follow-up survey conducted in Burkina Faso (+17.4 percentage points) [20]. This is consistent with findings that phone-based surveys overestimate contraceptive use compared to face-to-face interviews [43]. Other explanations for this difference between the two estimates are plausible, notably because of differences in timing and study populations [20]. Although our study assesses the effects of the COVID-19 pandemic one year after the onset of the pandemic, rather than three months, our estimate is within the 95% confidence interval (7.9–26.9) obtained in the Wood et al. study [20]. Importantly, findings from both studies converge and indicate no disruption to sexual and reproductive health services in Burkina Faso, and specifically, no shortages in contraceptives.

Nonetheless, women's sexual and reproductive health remains a concern. Indeed, the percentage of women with unwanted pregnancies increased, suggesting that demand for contraception has grown faster than access to it. Moreover, the rate of pregnancies that did not end with a live birth has also risen. Although it is impossible to distinguish between the different reasons (i.e., miscarriage, stillbirth, or abortion), the possibility that some of these were caused by SARS-CoV-2 infections in pregnant women cannot be excluded [15]. Other hypotheses include a higher rate of voluntary interruption of pregnancy, reduced use of (or access to) health facilities for antenatal care and deliveries, and poorer quality of care [44].

This study assessed the effects of COVID-19 on reproductive health indicators using a repeated pre-post survey in a panel of households. Most of the analyses used fixed effects regression at the individual level. This controls for any stable ("time-invariant") characteristics of the participants, whether observed or not, and allows to better isolate the effect of the COVID-19 pandemic. While the possibility of unobserved time-varying factors affecting the results cannot be ruled out, this evaluation design is one of the most rigorous in the absence of a control group [45, 46]. Unfortunately, without a control group, it is impossible to estimate (nor to control for) what would have been the "normal trend" between 2020 and 2021. Also, a social desirability bias in the responses is possible, given the sensitive nature of questions about reproductive health. However, the same questionnaire and data collection procedures were used for both surveys, so any information bias would likely have remained constant. Finally, this study has investigated the effects of the pandemic on four different outcomes. This is challenging, if only because some of these outcomes relate to rare events; the sample size was not determined by power calculation for the current analyses, since the panel was formed prior to the pandemic. More than a limitation, this constitutes one of this study's unique characteristics, allowing us to reach a more comprehensive understanding of the pandemic's impact on reproductive health, notably by associating recent past events, current practices, and future

intentions of women of reproductive age. However, it was not possible to simultaneously explore all the associations between these outcomes and the COVID-19, such as the potential mediation effect of history of adverse event on the desire for conception.

## Conclusion

This study indicates that, compared to before the COVID-19 pandemic, several indicators of women's reproductive health have changed in rural Burkina Faso. Notably, women are more likely to intend not to have any more children or to postpone subsequent pregnancies. Use of contraception has increased significantly, but not enough to avoid an increase in the rate of unwanted pregnancies. While the proportion of women with planned pregnancies remained stable, findings revealed an increase in the history of adverse events during recent pregnancy. By successfully increasing access to family planning services before and in the face of COVID-19, Burkina Faso has likely limited the extent of the deleterious effects on reproductive health in LMICs predicted by many at the time of the pandemic.

## Supporting information

**S1 Text. Questions regarding knowledge and perceptions towards COVID-19.**
(DOCX)

## Acknowledgments

We would like to acknowledge the communities in the study area who enabled this work to take place, all the participants, the health districts authorities, and the support from the Institut National de Démographie et de Santé and the Société d'Études et de Recherches en Santé Publique.

## Author Contributions

**Conceptualization:** Thomas Druetz, Frank Bicaba, Martine Shareck, David-Martin Milot, Abel Bicaba.

**Data curation:** Frank Bicaba, Alice Bila, Cheick Tiendrebeogo.

**Formal analysis:** Thomas Druetz, Sarah Cooper.

**Funding acquisition:** Thomas Druetz.

**Investigation:** Thomas Druetz, Frank Bicaba, Alice Bila, Martine Shareck, David-Martin Milot, Cheick Tiendrebeogo, Abel Bicaba.

**Methodology:** Thomas Druetz, Sarah Cooper.

**Project administration:** Thomas Druetz, Abel Bicaba.

**Supervision:** Thomas Druetz.

**Validation:** Thomas Druetz, Martine Shareck, David-Martin Milot, Abel Bicaba.

**Visualization:** Thomas Druetz.

**Writing – original draft:** Thomas Druetz, Sarah Cooper.

**Writing – review & editing:** Thomas Druetz, Frank Bicaba, Alice Bila, Martine Shareck, David-Martin Milot, Cheick Tiendrebeogo, Abel Bicaba.

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
