## [Decision Letter · Decision Letter 0]

15 Dec 2021

PGPH-D-21-00481

Change in childbearing intention, use of contraception, unwanted pregnancies, and related adverse events during the COVID-19 pandemic: Results from a panel study in rural Burkina Faso

Dear Dr. Druetz,

Thank you for submitting your manuscript to PLOS Global Public Health. After careful consideration, we feel that it has merit but does not fully meet PLOS Global Public Health’s publication criteria as it currently stands. Therefore, we invite you to submit a revised version of the manuscript that addresses the points raised during the review process.

We look forward to receiving your revised manuscript.

Kind regards,

Hermano Alexandre Lima Rocha

Academic Editor

Journal Requirements:

1. Please include additional information regarding the survey or questionnaire used in the study and ensure that you have provided sufficient details that others could replicate the analyses. For instance, if you developed a questionnaire as part of this study and it is not under a copyright more restrictive than CC-BY, please include a copy, in both the original language and English, as Supporting Information.

3. In the Methods, please clarify that participants provided oral consent. Please also state in the Methods:

- Why written consent could not be obtained

- Whether the Institutional Review Board (IRB) approved use of oral consent

- How oral consent was documented

For more information, please see our guidelines for human subjects research: https://journals.plos.org/plosone/s/submission-guidelines#loc-human-subjects-research

4. Please ensure you have thoroughly discussed any potential limitations of this study within the Discussion section, including the potential impact of confounding factors.

5. In the online submission form, you indicated that "All anonymized data used for this analysis can be made available by contacting the

corresponding author under reasonable request.". All PLOS journals now require all data underlying the findings described in their manuscript to be freely available to other researchers, either 1. In a public repository, 2. Within the manuscript itself, or 3. Uploaded as supplementary information.

6. Please provide us with a direct link to the base layer of the map used in Figure 1 and ensure this location is also included in the figure legend. 

Please note that, because all PLOS articles are published under a CC BY license (creativecommons.org/licenses/by/4.0/), we cannot publish proprietary maps such as Google Maps, Mapquest or other copyrighted maps. If your map was obtained from a copyrighted source please amend the figure so that the base map used is from an openly available source.

Please note that only the following CC BY licences are compatible with PLOS licence: CC BY 4.0, CC BY 2.0  and CC BY 3.0, meanwhile such licences as CC BY-ND 3.0 and others are not compatible due to additional restrictions. If you are unsure whether you can use a map or not, please do reach out and we will be able to help you. 

The following websites are good examples of where you can source open access or public domain maps:

7. Please ensure that the funders and grant numbers match between the Financial Disclosure field and the Funding Information tab in your submission form. Note that the funders must be provided in the same order in both places as well. Currently, MS holds a Tier 2 Canada Research Chair in Urban Health Equity and Young People information is not included in the Funding Information.

Additional Editor Comments (if provided):

Thank you for your fine submission. Please see below the reviewers comments.

Reviewers' comments:

Reviewer's Responses to Questions

**Comments to the Author**

1. Does this manuscript meet PLOS Global Public Health’s publication criteria? Is the manuscript technically sound, and do the data support the conclusions? The manuscript must describe methodologically and ethically rigorous research with conclusions that are appropriately drawn based on the data presented.

Reviewer #1: Yes

Reviewer #2: Yes

2. Has the statistical analysis been performed appropriately and rigorously?

Reviewer #1: I don't know

Reviewer #2: Yes

3. Have the authors made all data underlying the findings in their manuscript fully available (please refer to the Data Availability Statement at the start of the manuscript PDF file)?

Reviewer #1: No

Reviewer #2: Yes

4. Is the manuscript presented in an intelligible fashion and written in standard English?

Reviewer #1: Yes

Reviewer #2: Yes

5. Review Comments to the Author

Reviewer #1: The manuscript addresses the relevant issue of reproductive health in the context of LMIC's in times of COVID19. The text is clear and well written, as is the methodological design. The results are quite interesting, but presented in a way that seems inappropriate, as follows:

Table 1: Does not inform the population of women to whom the data refer to: if all women of reproductive age (N=814 + 87; 597 + 64), or if only sexually active women (N=814; 597 ), since the totals of the different variables show inconsistencies. In addition, comparative data on the 540 women who were actually surveyed at both times (2020 and 2021) should be presented. The title also lacks place and years, and does not clarify that the time lapse is 12 months.

Figure 2: The title lacks to indicate the effect measure, the sampled population and the study site. The text mentions adjusted OR, but does not say for which variables, while the figure suggests crude OR.

Figure 3: The x-axis is missing. Text mentions about significant and non-significant results, but the levels of significance are not presented. It is not clear whether the texts referring to figures 2 and 3 (lines 284-285 and 288-292) are comments or footnotes.

Figure 4: The level of significance of the difference between the curves is not shown. Title is also inappropriate.

Table 2 and 3: Similar formatting issues, already mentioned.

Finally, the results of the analyses, although coming from different regression models, should be presented in tables with a basic standardization. The figures should only serve to illustrate the results in the tables.

The discussion seemed appropriate. However, a possible limitation of the study would be that the date of the last birth and the occurrence of miscarriages, abortions and stillbirths were not evaluated as potential confounding factors in the association between the desired lapse of time until the next pregnancy and the COVID19 pandemic.

Reviewer #2: I would like to suggest that the authors explain how they arrived at the number of 700 households, based on which parameters, which epidemiological criteria for calculating the sample size of women of childbearing age, in the methods. Was it based on the prevalence of contraceptive use, for example?

Furthermore, in lines 209-210 – authors wrote that interviewers were trained for qualitative research, but the study is quantitative.

In lines 341-343, the authors discussed measures taken locally to expand the supply of contraceptives in this population. But I did not find data on the birth rate, which would be interesting to contextualize how the health system works in this area to assist family planning.

6. PLOS authors have the option to publish the peer review history of their article (what does this mean?). If published, this will include your full peer review and any attached files.

**Do you want your identity to be public for this peer review?** For information about this choice, including consent withdrawal, please see our Privacy Policy.

Reviewer #1: No

Reviewer #2: **Yes: **Rocha, Sabrina G M O

---

## [Decision Letter · Decision Letter 1]

2 Mar 2022

Change in childbearing intention, use of contraception, unwanted pregnancies, and related adverse events during the COVID-19 pandemic: Results from a panel study in rural Burkina Faso

PGPH-D-21-00481R1

Dear Dr Druetz,

We are pleased to inform you that your manuscript 'Change in childbearing intention, use of contraception, unwanted pregnancies, and related adverse events during the COVID-19 pandemic: Results from a panel study in rural Burkina Faso' has been provisionally accepted for publication in PLOS Global Public Health.

Best regards,

Hermano Alexandre Lima Rocha

Academic Editor

Congratulations on the work. We hope it will bring good impacts on public health.

Reviewer Comments (if any, and for reference):

Reviewer's Responses to Questions

**Comments to the Author**

1. If the authors have adequately addressed your comments raised in a previous round of review and you feel that this manuscript is now acceptable for publication, you may indicate that here to bypass the “Comments to the Author” section, enter your conflict of interest statement in the “Confidential to Editor” section, and submit your "Accept" recommendation.

Reviewer #2: All comments have been addressed

2. Does this manuscript meet PLOS Global Public Health’s publication criteria? Is the manuscript technically sound, and do the data support the conclusions? The manuscript must describe methodologically and ethically rigorous research with conclusions that are appropriately drawn based on the data presented.

Reviewer #2: Yes

3. Has the statistical analysis been performed appropriately and rigorously?

Reviewer #2: Yes

4. Have the authors made all data underlying the findings in their manuscript fully available (please refer to the Data Availability Statement at the start of the manuscript PDF file)?

Reviewer #2: (No Response)

5. Is the manuscript presented in an intelligible fashion and written in standard English?

Reviewer #2: (No Response)

6. Review Comments to the Author

Reviewer #2: (No Response)

7. PLOS authors have the option to publish the peer review history of their article (what does this mean?). If published, this will include your full peer review and any attached files.

**Do you want your identity to be public for this peer review?** For information about this choice, including consent withdrawal, please see our Privacy Policy.

Reviewer #2: No
